# Timing of Surgery and Social Determinants of Health Related to Pathologic Complete Response after Total Neoadjuvant Therapy for Rectal Adenocarcinoma: Retrospective Study of National Cancer Database

Megan Mai [1], Jodi Goldman [1], Duke Appiah [2], Ramzi Abdulrahman [1,3] , John Kidwell [1,4] and Zheng Shi [1,3,*]

[1] School of Medicine, Texas Tech University Health Sciences Center, Lubbock, TX 79430, USA
[2] Department of Public Health, School of Population and Public Health, Texas Tech University Health Sciences Center, Lubbock, TX 79430, USA
[3] Radiation Oncology Clinic, University Medical Center Cancer Center, Lubbock, TX 79430, USA
[4] Colorectal Surgery Clinic, University Medical Center Cancer Center, Lubbock, TX 79430, USA
[*] Correspondence: zheng.shi@ttuhsc.edu; Tel.: +1-806-775-8296

**Abstract:** Total neoadjuvant therapy (TNT) for rectal adenocarcinoma (RAC) involves multi-agent chemotherapy and radiation before definitive surgery. Previous studies of the rest period (time between radiation and surgery) and pathologic complete response (pCR) have produced mixed results. The objective of this study was to evaluate the relationship between the rest period and pCR. This study utilized the National Cancer Database (NCDB) to retrospectively analyze 5997 stage-appropriate RAC cases treated with TNT from 2016 to 2020. The overall pCR rate was 18.6%, with most patients undergoing induction chemotherapy followed by long-course chemoradiation (81.5%). Multivariable logistic regression models revealed a significant non-linear relationship between the rest period and pCR ($p = 0.033$), with optimal odds at 14.7–15.9 weeks post radiation (odds ratio: 1.49, 95% confidence interval: 1.13–1.98) when compared to 4.0 weeks. Medicaid, distance to the treatment facility, and community education were associated with decreased odds of pCR. Findings highlight the importance of a 15–16-week post-radiation surgery window for achieving pCR in RAC treated with TNT and socioeconomic factors influencing pCR rates. Findings also emphasize the need for clinical trials to incorporate detailed analyses of the rest period and social determinant of health to better guide clinical practice.

**Keywords:** rectal cancer; total neoadjuvant therapy; pathologic complete response

## 1. Introduction

Historically, the standard of care for locally advanced rectal adenocarcinoma has been neoadjuvant radiation or chemoradiation, surgery, and adjuvant chemotherapy. Although this standard had good control over local disease, distant recurrence remains problematic and many patients have difficulty tolerating adjuvant chemotherapy after major abdominal and pelvic surgery, which has led to the development of total neoadjuvant therapy (TNT) [1,2]. TNT is a treatment strategy for rectal adenocarcinoma (RAC) that includes a course of multi-agent chemotherapy and a separate course of radiation with or without concurrent chemotherapy before definitive surgery. In 2016, the National Comprehensive Cancer Network included TNT, using neoadjuvant induction chemotherapy, into its guidelines as an option for locally advanced rectal cancer [3]. Since then, many clinical trials support the safety, efficacy, and increased survival with TNT [2,4–7].

TNT is growing in use over the years, and it has become an important cancer therapeutic in the treatment of patients with RAC. One study of the National Cancer Database (NCDB) found that 34.6% of patients with locally advanced rectal cancer received TNT in 2020 [6]. With mounting clinical trial evidence supporting TNT, debate persists over

the optimal timing of surgery after completing TNT. Although a longer delay of surgery allows the primary tumor more time to respond to radiation, it also increases the risk of intraoperative fibrosis and regrowth of unresponsive disease. Before the TNT era, clinical trials of neoadjuvant radiation with and without concurrent single-agent chemotherapy found mixed results regarding the benefit of delaying surgery on pCR [8,9]. However, these clinical trials studied delay of surgery as a binary variable as opposed to a continuous variable. It is important to understand how the rest period (end of radiation to surgery) in TNT influences pCR in real-world clinical practice. No large database studies have yet investigated the specific impact of the rest period in the setting of TNT. Therefore, the objective of this study was to evaluate the relationship between the rest period and pCR using data from the NCDB.

## 2. Materials and Methods

### 2.1. Study Design

The NCDB is a joint project of the Commission on Cancer (CoC) of the American College of Surgeons and the American Cancer Society [10,11]. The CoC's NCDB and the hospitals participating in the CoC's NCDB are the source of the de-identified data used herein; they are not involved in the statistical validity or data analysis of the present study. The present study is a retrospective study of the NCDB. Patients aged 18 years and older, diagnosed with primary RAC between 2016 and 2020, who received a multi-agent course of chemotherapy and a separate course of radiation-based therapy targeted to the primary site followed by definitive surgery at a CoC-accredited facility were included in the current study (Figure A1). The NCDB defines a patient as receiving multi-agent or single-agent chemotherapy depending on which came first in their treatment regimen. Staging criteria were the American Joint Committee on Cancer 8th edition clinical stages T2–4, any N or T1, N1–2. All cases of metastatic disease at diagnosis (cM1) were excluded. All patients were started on TNT and received definitive surgery without adjuvant therapy. The NCDB records the type of surgery with limited details. 'Wedge or segmental resection; partial proctectomy', which is referred to as 'partial proctectomy', includes but is not limited to anterior resection, Hartmann's operation, low anterior resection, transsacral rectosig-moidectomy, and total mesorectal excision. 'Total proctectomy' includes but is not limited to abdominoperineal resection. TNT is defined as multi-agent neoadjuvant chemotherapy and radiation-based therapy with dates of treatment before definitive surgery. Radiation-based therapy includes short-course radiation (25 Gy in 5 fractions), long-course chemoradiation (45–50 Gy in 25–28 fractions), nonstandard radiation doses (i.e., those who started but did not complete short-course or long-course regimens for any reason), or unknown. Those with an unknown clinical stage ($n$ = 27,983), clinical stage IV ($n$ = 23,213), clinical stage T0 or T1N0 ($n$ = 6), unknown treatment dates ($n$ = 689), or an unknown pathologic stage ($n$ = 312) were excluded (Figure A1). For TNT-specific criteria, those who did not receive trimodal therapy ($n$ = 22,622), received any adjuvant therapy ($n$ = 10,628), or had an unclear order of induction vs. consolidative chemotherapy ($n$ = 6456) were also excluded. The main exposure variable was the rest period defined as weeks from the end of radiation to definitive surgery. The main outcome was pathologic complete response (pCR) with the outcome of interest defined as ypT0N0 on pathologic staging.

### 2.2. Measures

The NCDB collects several variables relevant to demographic factors, treatment facility, cancer diagnosis, stage of disease, treatment details, all-cause mortality, and follow-up time. Demographic variables include age at diagnosis, sex, and race/ethnicity. Other variables include socioeconomic status (income, living in a metro area, and primary payer for health insurance), distance traveled to treatment facility, and community education measure. The NCDB defines community education measure as the percentage of adults aged 25 or older in the patient's zip code who did not graduate from high school. Clinical variables include the year of cancer diagnosis, clinical stage, initial tumor size, type of treatment facility,

radiation regimen, tumor boost, type of surgery, and the Charlson–Deyo score, which is a weighted score of comorbid conditions.

### 2.3. Statistical Analysis

For descriptive statistics, Pearson's Chi-square test was used for comparisons among categorical variables while mean, standard deviation, and independent *t*-test were used for comparisons among continuous variables with normal distribution. For continuous variables without normal distribution, median, interquartile range (IQR), and Wilcoxon rank-sum test were used. Logistic regression was used to estimate odds ratios (OR) and 95% confidence intervals (CI) for the association between duration of rest period and pCR. Restricted cubic splines were used to evaluate potential non-linear relationship between durations of the rest period and pCR. All statistical analyses were performed using the statistical package Stata (StataCorp. 2021. Stata Statistical Software: Release 17. College Station, TX, USA: StataCorp LLC.) and SAS software version 9.4 (SAS Institute, Inc., Cary, NC, USA). Statistical significance was defined as a *p*-value of less than 0.050.

## 3. Results

### 3.1. Descriptive Statistics

There were 5997 patients included in the study. The mean age at diagnosis (standard deviation) was 56.5 (11.4) years old (Table 1). The majority of patients were males (62.3%), Non-Hispanic White (75.5%), were primarily covered by private insurance or managed care (57.6%), lived in a metropolitan or urban area (92.5%), and traveled no more than 30 miles to their treatment facility (65.5%) (Table 1). The most common treatment facility was at an academic/research program (38.2%) (Table 1). The most common income group comprised those earning USD 74,062 or more (33.7%) (Table 1). A greater proportion of patients originated from communities where the proportion of people who did not graduate high school was 5.0–9.0% (Table 1). A community education measure of >15.3% represents the least educated community whereas a community education measure of <5.0% represents the most educated community.

**Table 1.** Baseline characteristics of patients.

| | Overall (N = 5997) | No pCR (*n* = 4888) | pCR (*n* = 1109) | pCR Rate | *p*-Value |
|---|---|---|---|---|---|
| Age | | | | | |
| Mean (SD [1]) | 56.5 (11.4) | 56.6 (11.5) | 56.3 (11.2) | | 0.483 |
| 18–39 years old | 464 (7.7%) | 380 (7.8%) | 84 (7.6%) | 18.1% | 0.221 |
| 40–49 years old | 1089 (18.2%) | 879 (18.0%) | 210 (18.9%) | 19.3% | |
| 50–59 years old (Ref) | 1979 (33.0%) | 1622 (33.2%) | 357 (32.2%) | 18.0% | |
| 60–69 years old | 1705 (28.4%) | 1378 (28.2%) | 327 (29.5%) | 19.2% | |
| 70–79 years old | 669 (11.2%) | 546 (11.2%) | 123 (11.1%) | 18.4% | |
| 80+ years old | 91 (1.5%) | 83 (1.7%) | 8 (0.7%) | 0.7% | |
| Sex | | | | | |
| Male (Ref) | 3736 (62.3%) | 3220 (62.2%) | 516 (62.9%) | 13.8% | 0.689 |
| Female | 2261 (37.7%) | 1957 (37.8%) | 304 (37.1%) | 13.5% | |
| Race/Ethnicity | | | | | |
| Non-Hispanic White (Ref) | 4529 (75.5%) | 3682 (75.3%) | 847 (76.4%) | 18.7% | 0.871 |
| Hispanic/Latinx | 625 (10.4%) | 516 (10.6%) | 79 (9.8%) | 17.4% | |
| Non-Hispanic Black | 454 (7.6%) | 375 (7.7%) | 109 (7.1%) | 17.4% | |
| Non-Hispanic Asian/Pacific Islander | 286 (4.8%) | 230 (4.7%) | 56 (5.1%) | 19.6% | |
| Other/unknown | 103 (1.7%) | 85 (1.7%) | 18 (1.6%) | 17.5% | |

Table 1. *Cont.*

|  | Overall (N = 5997) | No pCR (n = 4888) | pCR (n = 1109) | pCR Rate | *p*-Value |
|---|---|---|---|---|---|
| **Community education measure** [2] |  |  |  |  |  |
| >15.3% | 993 (16.6%) | 836 (17.1%) | 157 (14.2%) | 15.8% | 0.011 |
| 9.1–15.2% | 1342 (22.4%) | 1109 (22.7%) | 233 (21.0%) | 17.4% |  |
| 5.0–9.0% | 1534 (25.6%) | 1239 (25.4%) | 295 (26.6%) | 19.2% |  |
| <5.0% (ref) | 1210 (20.2%) | 952 (19.5%) | 258 (23.3%) | 21.3% |  |
| Unknown | 918 (15.3%) | 752 (15.4%) | 952 (15.0%) | 18.1% |  |
| **Income** |  |  |  |  |  |
| <USD 46,277 | 749 (12.5%) | 629 (12.9%) | 120 (10.8%) | 16.0% | 0.107 |
| USD 46,277–USD 57,856 | 1015 (16.9%) | 824 (16.9%) | 191 (17.2%) | 18.8% |  |
| USD 57,856–USD 74,062 | 1282 (21.4%) | 1057 (21.6%) | 225 (20.3%) | 17.6% |  |
| USD 74,062 or more (Ref) | 2018 (33.7%) | 1612 (33.0%) | 406 (36.6%) | 20.1% |  |
| Unknown | 933 (15.6%) | 766 (15.7%) | 167 (15.1%) | 17.9% |  |
| **Year of Diagnosis** |  |  |  |  |  |
| 2016 (Ref) | 396 (6.6%) | 338 (6.1%) | 58 (9.8%) | 14.7% | 0.018 |
| 2017 | 518 (8.6%) | 439 (8.5%) | 79 (9.6%) | 15.3% |  |
| 2018 | 1339 (22.3%) | 1088 (21.9%) | 251 (25.1%) | 18.8% |  |
| 2019 | 1897 (31.6%) | 1552 (32.8%) | 345 (24.3%) | 18.2% |  |
| 2020 | 1847 (30.8%) | 1471 (30.7%) | 376 (31.2%) | 20.4% |  |
| **Facility type** |  |  |  |  |  |
| Academic/Research Program (Ref) | 2290 (38.2%) | 1874 (38.3%) | 416 (37.5%) | 18.2% | 0.842 |
| Comprehensive Community Cancer Program | 1757 (29.3%) | 1438 (29.4%) | 319 (28.8%) | 18.2% |  |
| Integrated Network Cancer Program | 1256 (20.9%) | 1012 (20.7%) | 244 (22.0%) | 19.4% |  |
| Community Cancer Program | 230 (3.8%) | 184 (3.8%) | 46 (4.2%) | 20.0% |  |
| Unknown | 464 (7.7%) | 380 (7.8%) | 84 (7.6%) | 18.1% |  |
| **Metro/urban/rural area** |  |  |  |  |  |
| Metro/urban (Ref) | 5549 (92.5%) | 4512 (92.3%) | 1037 (93.5%) | 18.7% | 0.279 |
| Rural | 253 (4.2%) | 209 (4.3%) | 44 (4.0%) | 17.4% |  |
| Unknown | 28 (3.3%) | 167 (3.4%) | 28 (2.5%) | 14.4% |  |
| **Distance traveled** |  |  |  |  |  |
| 0–30 miles (Ref) | 3925 (65.5%) | 3173 (64.9%) | 752 (67.8%) | 19.2% | 0.094 |
| More than 30 miles | 1197 (20.0%) | 1001 (20.5%) | 196 (17.7%) | 16.4% |  |
| Unknown | 875 (14.6%) | 714 (14.6% | 161 (14.5%) | 18.4% |  |
| **Primary payer** |  |  |  |  |  |
| Private insurance/managed care (Ref) | 3452 (57.6%) | 2783 (56.9%) | 669 (60.3%) | 19.4% | 0.019 |
| Medicare | 1495 (24.9%) | 1218 (24.9%) | 277 (25.0%) | 18.5% |  |
| Medicaid | 653 (10.9%) | 563 (11.5%) | 90 (8.1%) | 13.8% |  |
| Uninsured | 241 (4.0%) | 199 (4.1%) | 42 (3.8%) | 17.4% |  |
| Other/Unknown | 156 (2.6%) | 125 (2.6%) | 31 (2.8%) | 19.9% |  |

[1] SD: standard deviation; [2] Percent of adults aged 25 or older in the patient's zip code who did not graduate high school.

The number of new TNT cases increased from 396 in 2016 to 1847 in 2020 (Table 1). With it, the pCR rate also increased from 14.7% in 2016 to 20.4% in 2020. The overall pCR rate from 2016 to 2020 was 18.5%. The logistic regression model testing for a linear association between rest and pCR revealed that the odds of pCR were 1.00 (95% CI: 0.99–1.01), adjusted for order of chemotherapy and radiation dosage. However, restricted cubic splines analysis revealed a flexible, nonlinear, and negatively parabolic relationship between the rest period and the odds of pCR (test for nonlinearity, *p* = 0.033), adjusted for order of chemotherapy and radiation dosage. Rest periods of 4.1–26.9 weeks were significantly associated with elevated odds of pCR with the highest odds at approximately 15.3 weeks (OR = 1.49, CI: 1.13–1.98) when compared to 4.0 weeks (Figure 1). However, due to the plateau of ORs, rest periods of 14.7–15.9 weeks also produced a signif-

icant OR of 1.49 (Table A1). After 15.9 weeks, the odds of pCR decreased, but remained statistically significant until 26.9 weeks.

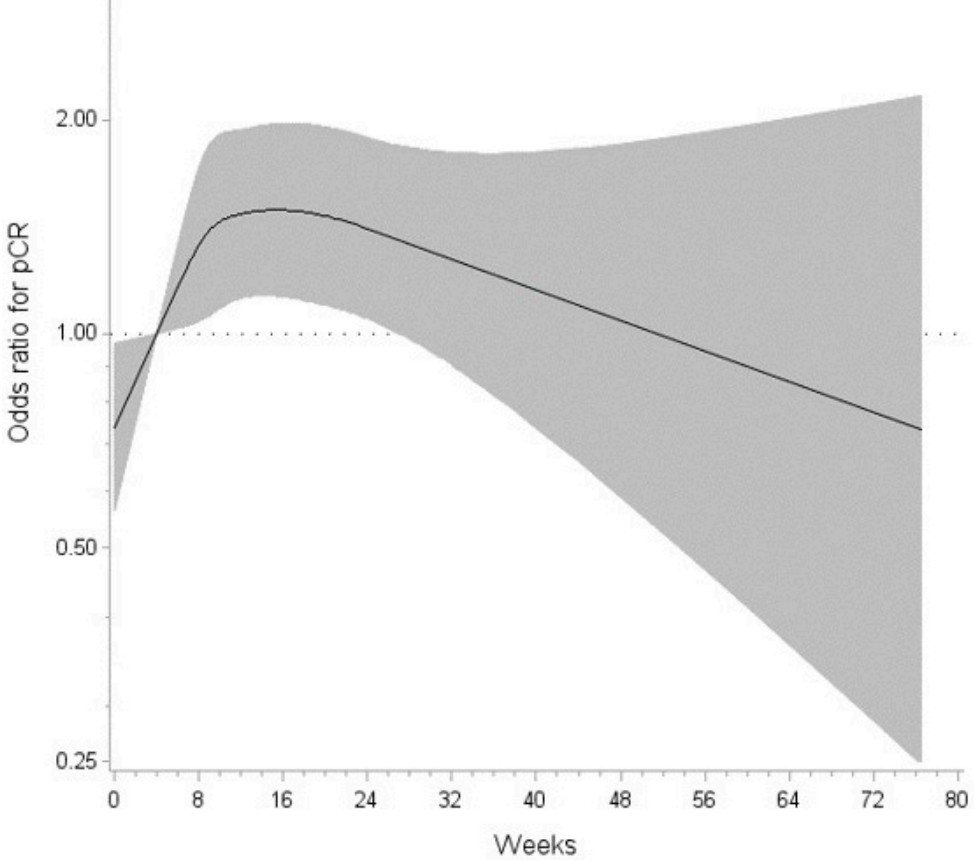

**Figure 1.** The relationship between the rest period and the odds ratio for pCR is non-linear (*p* = 0.033). Rest periods of 4.1–26.6 weeks were significantly associated with elevated odds of pCR with the highest odds estimated at 14.7–15.9 weeks (OR 1.49) when compared to 4.0 weeks. The gray shading represents the 95% confidence interval.

The majority of patients presented with clinical stage III (81.8%) and Charlson–Deyo score 0 (80.3%) (Table 2). A Charlson–Deyo score of score of 0 indicates 'no comorbid conditions recorded', or none of the values listed in the NCDB PUF data dictionary. The median initial tumor size (IQR) was 50 mm (35–66 mm). The predominant TNT regimen used in this cohort was induction chemotherapy followed by long-course chemoradiation (81.5%). The most common definitive surgery type was 'partial proctectomy' (64.9%), which includes but is not limited to anterior resection, Hartmann's operation, low anterior resection, transsacral rectosigmoidectomy, and total mesorectal exision. The median rest period (IQR) was 69 days (57–88).

**Table 2.** Clinical characteristics of patients.

| | Overall (N = 5997) | No pCR (*n* = 4888) | pCR (*n* = 1109) | pCR Rate | *p*-Value |
|---|---|---|---|---|---|
| **Clinical Stage** | | | | | |
| Stage I (T2N0M0) | 63 (1.1%) | 48 (1.0%) | 15 (1.4%) | 23.8% | 0.547 |
| Stage II | 1027 (17.1%) | 839 (17.2%) | 188 (17.0%) | 18.3% | |
| Stage III (Ref) | 4907 (81.8%) | 4001 (81.9%) | 906 (81.7%) | 18.5% | |
| **Initial tumor size (mm)** | | | | | |
| Median (IQR) | 50 (35–66) | 50 (35–66) | 50 (38–66) | | 0.48 |
| **Charlson–Deyo score** | | | | | |
| 0 (Ref) | 4815 (80.3%) | 3916 (80.1%) | 899 (81.1%) | 18.7% | 0.473 |
| 1 or more | 1182 (19.7%) | 972 (19.9%) | 210 (19.0%) | 17.8% | |
| **Order of chemotherapy** | | | | | |
| Induction chemotherapy (Ref) | 5177 (86.3%) | 4221 (86.2%) | 966 (87.1%) | 18.7% | 0.403 |
| Consolidative chemotherapy | 820 (13.7%) | 677 (13.9%) | 143 (12.9%) | 17.4% | |
| **Radiation dosing** | | | | | |
| 25 Gy in 5 fx | 538 (9.0%) | 427 (8.7%) | 111 (10.0%) | 20.65 | 0.216 |
| 45–50 Gy in 25–28 fx | 4597 (76.7%) | 3740 (76.5%) | 857 (77.3%) | 18.6% | |
| Nonstandard dose (Ref) | 605 (10.1%) | 504 (10.3%) | 101 (9.1%) | 16.7% | |
| Unknown | 257 (4.3%) | 217 (4.4%) | 40 (3.6%) | 15.6% | |
| **TNT regimen** | | | | | |
| I-chemotherapy + L-XRT [1] (Ref) | 4184 (81.5%) | 3389 (81.0%) | 795 (19.0%) | 19.0% | 0.107 |
| L-XRT + C-chemotherapy [2] | 413 (8.0%) | 351 (85.0%) | 62 (15.0%) | 15.0% | |
| S-XRT + C-chemotherapy [3] | 289 (5.6%) | 225 (77.9%) | 64 (22.2%) | 22.2% | |
| I-chemotherapy + S-XRT [4] | 249 (4.9%) | 202 (81.1%) | 47 (18.9%) | 18.9% | |
| **Tumor boost** | | | | | |
| None or incomplete boost (Ref) | 3261 (54.4%) | 2670 (54.6%) | 591 (53.3%) | 18.1% | 0.074 |
| 5.4 Gy in 3 fx | 2485 (41.4%) | 2002 (41.0%) | 483 (43.6%) | 19.4% | |
| Unknown | 251 (4.2%) | 216 (4.4%) | 35 (3.2%) | 13.9% | |
| **Type of surgery** | | | | | |
| Total proctectomy (Ref) | 1432 (23.9%) | 1211 (24.8%) | 221 (19.9%) | 15.4% | <0.001 |
| Partial proctectomy | 3893 (64.9%) | 3108 (63.6%) | 785 (70.8%) | 20.2% | |
| Pull through with sphincter preservation | 381 (6.4%) | 304 (6.2%) | 77 (6.9%) | 20.2% | |
| Other proctectomy, unspecified | 291 (4.9%) | 265 (5.4%) | 26 (2.3%) | 8.9% | |

[1] I-chemotherapy (Induction chemotherapy); [2] L-XRT (Long-course XRT, 45–50 Gy in 25–28 fx); [3] C-chemotherapy (Consolidation chemotherapy); [4] S-XRT (Short-course XRT, 25 Gy in 5 fx).

*3.2. Factors Associated with pCR*

Of the 17 factors relevant to social determinants of health and clinical characteristics that were evaluated, 5 were statistically significantly associated with pCR outcomes in multivariable analysis: year of diagnosis, living in a zip code with a lower-education population, traveling more than 30 miles to the treatment facility, having Medicaid as the primary payor, and type of surgery. On the one hand, patients diagnosed with RAC in 2020 had higher odds of pCR compared to those diagnosed in 2016 (OR: 1.49; 95% CI 1.10–2.01) as well as those who had partial proctectomy (OR: 1.38; 95% CI 1.18–1.63). On the other hand, the odds for pCR were lower in patients who lived in zip codes with fewer proportions of adults (<9.0%) who did not graduate high school (OR: 0.79; 95% CI 0.69–0.92), patients who travelled more than 30 miles to the treatment facility (OR: 0.83, 95% CI 0.70–0.98) and those whose mode of payment was Medicaid compared to private insurance/managed care (OR: 0.66; 95% CI 0.52–0.84).

## 4. Discussion

In the current study, a significant nonlinear relationship was observed between rest period and pCR after adjusting for potential confounding factors including the order of chemotherapy and radiation dosage. Specifically, a rest period of 14.7 to 15.9 weeks was observed to be an ideal window prior to surgery to achieve pCR after TNT in rectal cancer as this period was associated with 50% higher odds of pCR in a study sample with 81.5% of patients receiving induction chemotherapy with long-course chemoradiation. After 15.9 weeks, the odds of pCR decrease, but remain statistically significant until 26.9 weeks (Figure 1). Further decrease in pCR after 26.9 weeks is difficult to interpret and could be attributed to cases of watch and wait followed by salvage surgery or cases with unfavorable tumor biology; however, these data are not collected by the NCDB. Of note, this association was independent of consolidation vs. induction chemotherapy and short- vs. long-course radiation. This is the first comprehensive quantitative analysis of the timing of surgery to achieve pCR post-neoadjuvant radiation which contributes to rectal cancer management in the TNT era.

Previous studies have evaluated the role of the rest period and pCR, but none have analyzed it with cubic restricted splines. One of the landmark TNT trials, CAO/ARO/AIO-12, found the group that received consolidative chemotherapy achieved an increased pCR rate (25%, $p < 0.001$) relative to the historical standard of care (15%), whereas the induction chemotherapy group failed to achieve a statistically significant increase in pCR (17%, $p = 0.210$) relative to historical standards [4]. Those receiving consolidation chemotherapy had a median of 90 days from end of chemoradiation to surgery, or 'rest period.' Those receiving induction chemotherapy had a median of 45-day 'rest period'. Of note, there was no statistical difference in 3-year disease-free survival or 3-year cumulative incidence of locoregional recurrence after a median follow-up time of 43 months. In accordance with the results of the current study, the increase in pCR for the group receiving consolidative chemotherapy in the CAO/ARO/AIO-12 trial may be related more to a longer rest period between chemoradiation and surgery than the actual order of multiagent chemotherapy to chemoradiation [5].

Although the TNT trials were not designed to specifically study the rest period, differences in treatment regimen allow for variable durations of the rest period. For example, the rectal cancer and preoperative induction therapy followed by a dedicated operation (RAPIDO) trial investigated safety and efficacy of short-course radiation, consolidative CAPOX or FOLFOX, surgery vs. standard of care, which consisted of neoadjuvant long-course chemoradiation with concurrent capecitabine, surgery, with or without adjuvant CAPOX or FOLFOX [6]. Of note, the rest period in the TNT arm lasted at least 22 weeks due to a course of consolidative chemotherapy vs. 6–10 weeks in the standard of care arm. The pCR rate in the TNT arm was 28% vs. 14% in the standard of care arm. The increased pCR rate observed in the TNT arm could partially be attributed to a longer rest period. However, a longer rest period is not the only factor that can contribute to increased pCR rates. The UNICANCER-PRODIGE 23 trial controlled for the rest period by studying neoadjuvant fluorouracil, leucovorin, irinotecan, and oxaliplatin, neoadjuvant long-course chemoradiation, surgery vs. standard of care, which consisted of neoadjuvant CAPOX or FOLFOX, surgery, absence or presence of adjuvant CAPOX or FOLFOX [2]. Despite having a study protocol that dictated surgery 6–8 weeks post-radiation completion in both the TNT arm and the standard-of-care arm, the results revealed a pCR rate of 28% in the TNT arm vs. 12% in the standard-of-care arm, indicating that factors other than rest period also increase the odds of pCR.

In most of the TNT trials, the focus of the protocols has been on the duration from the end of TNT to surgery [4,6,7]. Our study adds to the literature by emphasizing the importance of the period from the end of radiation to surgery regardless of whether the patient receives induction multi-agent chemotherapy or consolidative multi-agent chemotherapy, which is largely consistent with previous clinical trials. Although our study's primary end-point is pCR, we recognize the prognosis of patients with and without

pCR remains requires high-quality longitudinal data and detailed survival analysis that differentiates overall survival from cancer-specific survival, which are beyond the scope of this study.

To date, a deficit in the literature exists in guiding clinicians on the most appropriate length of the rest period between completing TNT and surgery. Recent clinical trials have begun to address this topic by studying 'watchful waiting' periods. The Organ Preservation in Patients with Rectal Adenocarcinoma trial compared TNT using induction to consolidative neoadjuvant chemotherapy with fluorouracil, leucovorin and oxaliplatin (FOLFOX) or capecitabine and oxaliplatin (CAPOX) as neoadjuvant chemotherapy and fluorouracil or capecitabine-based chemoradiation [7]. Roughly 70% of patients reached complete or near-complete clinical response and were offered a watchful waiting period with strict surveillance protocols. Of note, there was no significant difference in disease-free survival for patients who underwent watchful waiting compared to those receiving planned total mesorectal excision following neoadjuvant therapy at the restaging point as per standard of care. The median watchful waiting period, from restaging to cancer regrowth and total mesorectal excision, was 30 weeks [6]. These findings are consistent with the current results that show non-significant odds of pCR approximately 27.0 weeks post-radiation completion. This clinical trial highlights the importance of re-evaluation after TNT and before surgery to determine whether a patient is a good candidate for watchful waiting and the difficulty of timing surgery if a patient shows incomplete clinical response at the time of re-evaluation. In another retrospective study, a rest period of more than 90 days was associated with increased radial margin positivity, presumably attributed to intraoperative fibrosis and increased operative difficulty [12]. In clinical practice, the risks of increased operative difficulty must be weighed against the benefits of pCR and organ preservation when considering the timing of surgery. While evidence-based medicine is important for improving overall cancer outcomes, multidisciplinary tumor boards are equally important for individualized treatment planning.

Although social determinants of health were not the main focus of this study, further discussion is relevant to the holistic treatment of RAC. Many retrospective studies have shown that patients who have Medicaid or no insurance encounter many barriers that contribute to worse cancer outcomes, including delays in treatment, differences in treatment received, and transportation issues [13,14]. There were three socioeconomic factors that significantly decreased the odds of pCR in our study: Medicaid, traveling longer distances to the treatment facility, and living in a zip code with lower community education. The NCDB measures community education as the percentage of adults aged 25 or older in the patient's zip code who did not graduate from high school. Those who lived in zip codes with the lowest percentage of high school graduates by age 25 had lower odds of pCR. It is important to note that this variable is not a measure of the individual's education, but rather reflective of the local community. Interestingly, Medicaid patients had a significantly lower pCR rate even when compared to uninsured patients. Although the prevalence of Medicaid users and uninsured patients in this study was relatively small, these findings are consistent with other retrospective studies. Our study also showed that traveling more than 30 miles to the treatment facility was associated with lower pCR rates. These findings highlight the importance of minimizing barriers to cancer treatment in real-world practice.

With a view into sizable real-world cohorts, these results are novel and important because of the consideration of a nonlinear relationship between the rest period and odds of pCR using restricted cubic splines analysis which provides a better characterization of the complex relationship between the rest period and pCR rates compared to previous studies that evaluated the rest period as a categorical construct. The cubic spline model offers accurate information about the changes in the OR of pCR over time relative to the reference point. Another strength of this study is the large number of rectal adenocarcinoma cases with information regarding the timing of radiation and surgery. Although national databases are strategic for studying large sample sizes, TNT regimens differ across institutions (short- vs. long-course radiation; double- vs. triple-agent chemotherapy). Despite

a heterogenous study sample, these findings are clinically relevant when coordinating multimodal treatment regimens.

The NCDB provides an opportunity to uncover answers to questions that are unlikely to be studied in a prospective randomized clinical trial but does present several limitations including selection bias and missing data for important variables. For example, the NCDB does not collect information about tolerability, adverse treatment-related events, and completion rate of TNT. Furthermore, the NCDB is not optimized for collecting TNT-specific data. In particular, a large majority of the sample received induction chemotherapy (86.3%) vs. consolidation chemotherapy (13.7%). Only one variable is available to report, if a patient received single- vs. multi-agent chemotherapy, and whichever comes first in a patient's treatment regimen is what is listed. A patient may be listed as receiving single-agent chemotherapy if they received a single-agent alongside radiation followed by multi-agent chemotherapy. Although this regimen fits within the criteria for TNT, this patient would have been excluded as it was not possible to determine if such patient did in fact receive multi-agent chemotherapy as it was not recorded. Our study sample is highly selective due to the proportion of cases excluded for missing information. We used restriction instead of multiple imputation as these missing data were not assumed to be missing at random. Further prospective studies can be designed to minimize missing information. Lastly, the NCDB is inherently characterized by selection bias due to participation of CoC-accredited facilities across the United States but is not necessarily generalizable to the national population due to limited participation of facilities.

## 5. Conclusions

The current NCDB analysis demonstrated that the period from 14.7 to 15.9 weeks within completing the radiation-based portion of TNT is an ideal window to achieve pCR in RAC when surgery is performed. However, 4.1 to 26.9 weeks between the end of radiation and surgery is an adequate time frame to achieve pCR after TNT in RAC. Other factors, such as Medicaid coverage, long travel distances, and lower community education, also impacted pCR rates. Considering the variability of treatment timing and regimen across the NCDB cohort, prospective studies of TNT are warranted to further study rest period and pCR rates using consistent therapy strategies.

**Author Contributions:** Conceptualization, M.M., Z.S. and D.A.; methodology, M.M., Z.S. and D.A.; formal analysis, M.M. and D.A.; investigation, M.M., Z.S. and D.A.; resources, M.M., Z.S. and D.A.; data curation, M.M. and D.A.; writing—original draft preparation, M.M., Z.S., D.A., J.G., R.A. and J.K.; writing—review and editing, M.M., Z.S., D.A., J.G., R.A. and J.K.; visualization, M.M. and Z.S.; supervision, Z.S. All authors have read and agreed to the published version of the manuscript.

**Funding:** This research received no external funding.

**Institutional Review Board Statement:** This project was determined to represent research that is exempt from formal review by the institutional review board by the Texas Tech University Health Sciences Center Institutional Review Board (IRB#: L22-227). The study was found not to require formal IRB review because the research falls into one of the categories specifically designated as exempt per 45 CFR 46.104(d).

**Informed Consent Statement:** Not applicable.

**Data Availability Statement:** Data used for this study were only available to facilities accredited by the Commission Cancer of the American College of Surgeons and American Cancer Society.

**Conflicts of Interest:** The authors declare no conflicts of interest.

**Appendix A**

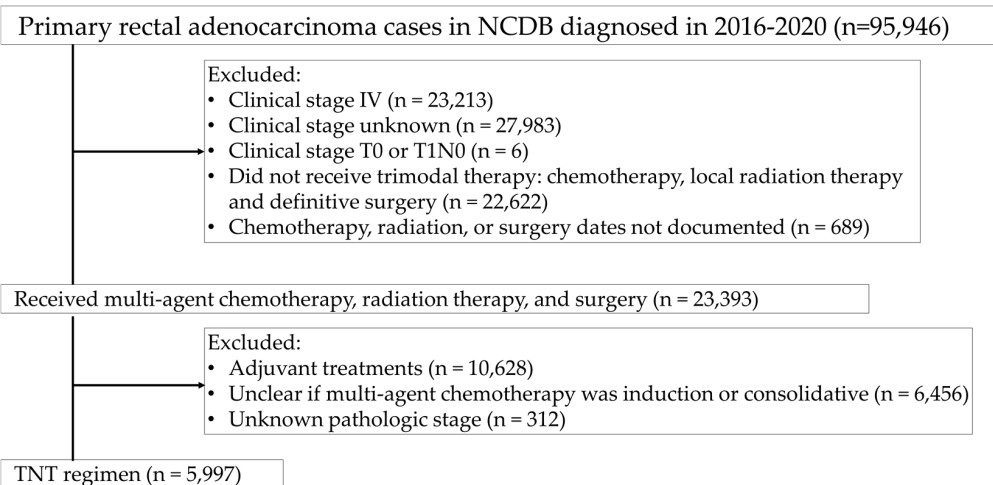

Excluded:
- Clinical stage IV (n = 23,213)
- Clinical stage unknown (n = 27,983)
- Clinical stage T0 or T1N0 (n = 6)
- Did not receive trimodal therapy: chemotherapy, local radiation therapy and definitive surgery (n = 22,622)
- Chemotherapy, radiation, or surgery dates not documented (n = 689)

Primary rectal adenocarcinoma cases in NCDB diagnosed in 2016-2020 (n=95,946)

Received multi-agent chemotherapy, radiation therapy, and surgery (n = 23,393)

Excluded:
- Adjuvant treatments (n = 10,628)
- Unclear if multi-agent chemotherapy was induction or consolidative (n = 6,456)
- Unknown pathologic stage (n = 312)

TNT regimen (n = 5,997)

**Figure A1.** Study design flow chart.

**Table A1.** Restricted cubic splines for the association of the rest period and pCR.

| Rest Period (Weeks) | OR | (95% CI) |
|---|---|---|
| 0.1531428528 | 0.747 | (0.577, 0.968) |
| 4.0 (ref) | 1.000 | (1.000, 1.000) |
| 4.1348570251 | 1.010 | (1.001, 1.019) |
| 14.701713867 | 1.494 | (1.132, 1.972) |
| 14.85485672 | 1.494 | (1.131, 1.973) |
| 15.007999573 | 1.494 | (1.131, 1.974) |
| 15.161142426 | 1.494 | (1.131, 1.975) |
| 15.314285278 | 1.494 | (1.130, 1.976) |
| 15.467428131 | 1.494 | (1.129, 1.977) |
| 15.620570984 | 1.494 | (1.129, 1.978) |
| 15.773713837 | 1.494 | (1.128, 1.978) |
| 15.926856689 | 1.494 | (1.127, 1.979) |
| 26.95314209 | 1.357 | (1.001, 1.840) |
| 76.571426392 | 0.733 | (0.249, 2.159) |

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
