# Peer review of "Timing of Surgery and Social Determinants of Health Related to Pathologic Complete Response after Total Neoadjuvant Therapy for Rectal Adenocarcinoma: Retrospective Study of National Cancer Database"

_curroncol, doi:10.3390/curroncol31030097_

Round 1

Reviewer 1 Report

Comments and Suggestions for Authors

General comments:

The authors reported a retrospective study using the NCDB that investigated the relationship between “rest period” and pCR after curative surgery followed by TNT for rectal cancer. They found that rest period of 15-16 weeks post radiation showed optimal results with 1.5 odds ratio compared to 4 weeks, and it was shown using a unique restricted cubic spline model. Interestingly, this association was independent of induction vs. consolidation chemotherapy and short- vs. long-course radiation.

They also found that year of diagnosis (2020 vs. 2016) and type of surgery (partial proctectomy) showed higher odds of pCR: while patients living in a zip code with a lower education, traveling more than 30 miles to the treatment facility, and patients having Medicaid as the primary payor showed lower odds of pCR.

The results were deemed to reflect real-world data with diverse treatment timings and regimens in a mixed medical and socioeconomical society. This study clearly shows the optimal timing of surgery to achieve pCR after TNT, using universally available database.

Specific comments:

Despite the unique results and helpful recommendations, many limitations do exist in this study. The discussion may be extended considering the following points:

1.       Tolerability (adverse events) and completion rate of TNT were not shown.

2.       Prognosis of the patients with pCR and non-pCR remains unknown.

3.       The type of surgery, “partial proctectomy” including wedge or segmental resection, partial proctectomy, anterior resection, Hartmann’s operation, low anterior resection, and transsacral rectosigmoidectomy and total mesorectal excision, had more favorable pCR rate over “total proctectomy” including but not limited to abdominoperineal resection. This was due to the limited information on the NCBD records, but the results may appear confusing in the era of achieved conceptualization and technical standardization of total mesorectal excision in rectal cancer surgery.

Reviewer 2 Report

Comments and Suggestions for Authors

The paper is well written. The results, especially in terms of SDH are important.

This study will be cited frequenctly.

Specific comments:

The main question addressed by the research are:

Association of pCR with SDOH (social determinants of health); in terms of medicaid, distance from tmt, community education.

The SDOH are all the rage now that CMS is forcing them to be recorded. They are ahead of the wave.

This paper does an excellent to associate SDOH with pCR.

This article introduces about new knowledge regards SDOH and pCR.

The authors just need to be clear that they are using medicaid status, distance from tmt and education as surrogates for SDOH. The measures say socioeconomic status while the title says SDOH. Maybe the measures is where they could elaborate why their measures are SDOH?

The references are appropriate.

Tables good. Figure 1 looks like final result can be anywhere, maybe because 0.25 on y-axiz and 0 on x-axis. Maybe make OR 0-3, cut out the 3 to 4, and explain the trombone like shading.

Reviewer 3 Report

Comments and Suggestions for Authors

Thanks for submitting our manuscript. 

I have questions regarding the analysis of optimal rest period. 

Although the optimal rest period suggested is 14.7-15.9 wks based on an OR of 1.49 this is compared to only the 4 wk mark which most would consider too soon to see benefit of radiation.

Most surgeons (and trials) suggest an 8-10 week rest period from end of radiation to surgery is ideal as waiting longer increases risk of fibrosis and difficulty in surgery. What is the OR of pCR for 14.7 - 15.9 wks compared to 8-10 wks? Is it significant?

Although issues of fibrosis alluded to in the introduction, I would suggest some discussion of this in the discussion part. This may be a limitation of suggesting the optimal rest period above.

Thanks

Lloyd
